# Microbiota Modulation in Patients with Metabolic Syndrome

**DOI:** 10.3390/nu14214490

**Published:** 2022-10-25

**Authors:** Ricardo Araujo, Marta Borges-Canha, Pedro Pimentel-Nunes

**Affiliations:** 1Nephrology & Infectious Diseases R&D Group, i3S—Instituto de Investigação e Inovação em Saúde, Universidade do Porto, 4200-135 Porto, Portugal; 2INEB—Instituto de Engenharia Biomédica, Universidade do Porto, 4200-135 Porto, Portugal; 3Department of Surgery and Physiology, Faculty of Medicine, University of Porto, 4200-319 Porto, Portugal; 4Department of Endocrinology, Diabetes and Metabolism, Centro Hospitalar Universitário de São João, 4200-319 Porto, Portugal; 5RISE@CI-IPOP (Health Research Network, IPO Porto), Porto Comprehensive Cancer Center (Porto CCC), 4200-072 Porto, Portugal

**Keywords:** inflammation, gut metabolites, gut microbiome, metabolic syndrome, obesity, probiotics, synbiotics

## Abstract

Metabolic syndrome (MS) comprises a vast range of metabolic dysfunctions, which can be associated to cardiovascular disease risk factors. MS is reaching pandemic levels worldwide and it currently affects around 25% in the adult population of developed countries. The definition states for the diagnosis of MS may be clear, but it is also relevant to interpret the patient data and realize whether similar criteria were used by different clinicians. The different criteria explain, at least in part, the controversies on the theme. Several studies are presently focusing on the microbiota changes according to the components of MS. It is widely accepted that the gut microbiota is a regulator of metabolic homeostasis, being the gut microbiome in MS described as dysbiotic and certain taxonomic groups associated to metabolic changes. Probiotics, and more recently synbiotics, arise as promising therapeutic alternatives that can mitigate some metabolic disturbances, namely by correcting the microbiome and bringing homeostasis to the gut. The most recent studies were revised and the promising results and perspectives revealed in this review.

## 1. Introduction

Metabolic syndrome (MS), also known as syndrome X or insulin resistance syndrome, comprises a constellation of metabolic dysfunctions, which represent cardiovascular (CV) disease risk factors [1]. Its definition may be controversial according to various entities [1]. One of the most accepted and used definition is the one recommended by the National Cholesterol Education Program (NCEP), 2005 [2]. The definition states that the diagnosis may be made in the presence of any three or more of the following: (1) fasting blood glucose greater than 100 mg/dL or drug treatment for elevated blood glucose; (2) high-density lipoprotein (HDL) cholesterol <140 mg/dL in men or <50 mg/dL in women, or drug treatment for low HDL cholesterol; (3) blood triglycerides > 150 mg/dL or drug treatment for elevated triglycerides; (4) waist circumference > 102 cm in men or >88 cm in women; (5) blood pressure > 130/85 mmHg or drug treatment for hypertension [2]. When interpreting data, it is important to realize whether this or other criteria were used by the authors. The different criteria used in the existing literature explain, at least in part, the controversies on this theme.

The complex and not entirely clear pathophysiology of MS is largely acknowledged [1]. Abdominal adiposity and insulin resistance are thought to be central elements for its development [3]. Data shows complex interactions between internal factors, as genetic backgrounds, as well as external factors, such as physical activity and diet [4,5]. Nonetheless, genetic background is believed to be only a minor component for MS development, given the epidemic grow of such metabolic disturbance, which is unlikely related to genetics [6]. On the other hand, epigenetic changes namely in the spermatozoa, oocytes or in utero may have an important role [1]. Nutrition (both intrauterine and postnatal) and growth have also shown strong associations with MS in the adulthood [1]. Inflammation may also be an important contributing factor to the metabolic dysfunction [7]. This led to the concept of immunometabolism, linking inflammation, and metabolic defects [7,8]. For instance, MS is now known for being a milieu of a chronic pro-inflammatory state namely presenting with elevated inflammatory cytokines (such as tumour necrosis factor-α and interleukin-6) and acute-phase reactants (such as C-reactive protein and fibrinogen) [5]. Data shows that inflammatory cytokines associated to MS stimulate insulin resistance in adipose tissue and muscle [5].

MS is reaching pandemic levels worldwide and it currently affects around 25% in the adult population of developed countries [1]. The rising prevalence of MS parallels obesity and type 2 diabetes prevalence’s, which are often coincidental [1]. Identifying these patients is crucial to achieve their optimal CV risk management. MS components are independent risk factors for CV disease and the combination of them may be synergic [9]. Given the uncertainty on its pathophysiology and aetiology, as well as the great variability among different individuals, the best treatment approach is not known [3]. It is consensual that prevention rather than treating should be the targeted, and that no single medication can eradicate it [1]. Currently, lifestyle changes (namely concerning diet and exercise) are basilar in the treatment of patients with MS [10]. Different recommendations are available and most include the goal of 7-10% weight loss, regular moderate intensity physical activity (according to the patient’s clinical status) and adopting a diet with low intake of saturated fat, transfat, and cholesterol [5]. Individual pharmacological therapy may address central adiposity, insulin resistance, dyslipidaemia, hypertension, and hypercoagulable state [3]. Additionally, in the setting of severe obesity, bariatric surgery is a greatly effective treatment of multiple risk factors [3].

In this review we will explore the relationship between MS and the gut microbiome and the potential of microbial modulators (probiotics or synbiotics) to interfere with the disease and improve patients’ health. In addition, a systematic review on the randomized control trials conducted using probiotics or synbiotics in patients with MS will be shown.

## 2. Metabolic Syndrome and Microbiota

The human gut is known for its wide microbiota composition, which usually lives in a symbiotic relationship with the host. These microorganisms use the undigested nutrients reaching the colon as substrates to live, and some of the microbes are important to final product degradation and for vitamin formation, among other crucial functions related to host’s immunity [11,12,13].

It is widely accepted that the gut microbiota is a regulator of metabolic homeostasis [14,15,16,17]. Particularly, multiple latest studies aimed to characterize the role of the microbiota in the pathogenesis of MS, given that these two are thought to be highly correlated. Although the specific microorganism profile in patients with MS is not yet known, it seems likely that these patients have a different microbiota composition (dysbiosis), when compared to patients without MS (Figure 1). This different milieu, including different bacterial metabolites, may regulate inflammation and immunity, as well as the metabolic homeostasis [18]. The recognition of the microbiome impact on metabolism is recent and yet to be elucidated. Possible explanations for this regulation, which likely act together, may embrace the regulation by the microbiome of epithelial lipid uptake, hepatic gluconeogenesis, circadian host biology, and insulin signalling, among other possible mechanisms [17].

Concerning the microbiome profile, the HELIUS study, a multi-ethnic population study, reported higher proportion of Enterobacteriaceae and lower of Peptostreptococcaceae in patients with MS [19]. Also, enrichment of Enterobacteriaceae, as well as in *Turicibacter* sp., *Clostridium coccoides, Clostridium leptum*, and decrease of *Butyricicoccus* sp., *Akkermansia muciniphila*, and *Faecalibacterium prausnitzii* was reported in Romanian patients with MS [20]. Similarly, Qin and colleagues [21] reported microbiota changes in patients with MS namely decreased abundance of *Alistipes onderdonkii, Clostridium asparagiforme, Clostridium citroniae, Clostridium scindens, Roseburia intestinalis*, and *Bacteroides thetaiotaomicron*. Walker and colleagues [22] performed a population cross-sectional analysis in which from the 8 operational taxonomic units (OTUs) associated with diabetes, 3 OTUs (identified as belonging to Ruminococcaceae, Clostridiales, and Lachnospiraceae) were also significantly associated with MS and CV disease risk. These results advocate that microbiota may mediate mechanisms that contribute to cardiometabolic phenotypes through common mechanisms.

There are also studies focusing on the microbiota changes according to the components of MS. For example, Atzeni and colleagues [23] aimed to determine different faecal microbiota signatures associated with insulin resistance in a population with MS and concluded that differences in insulin resistance associated to a singular microbiota profile. These authors reported a negative association between insulin resistance and *Desulfovibrio, Odoribacter,* and Oscillospiraceae UCG-002, through mechanism of amino acid degradation, gluconeogenesis, immunomodulation and acetate, and a positive association between insulin resistance and *Feacalibaterium* and *Butyricicoccus* linked with the production of butyrate [23].

Yan and colleagues [24] studied 41 patients to identify gut microbiota changes in patients with visceral obesity. These authors found strong correlations between 16 species and visceral adiposity, being the strongest one with *Escherichia coli*. Additionally, the degradation of short-chain fatty acids (SCFAs) may be related to visceral adipose accumulation. The authors underline the hypothesis of an intrinsic connection between the gut microbiota and visceral adiposity, as well as the related metabolic disorders.

The METISM cohort is a Finland population cohort composed by unrelated man primarily designed to determine the prevalence and genetic determinants of metabolic and CV diseases. Org and colleagues [25] aimed to investigate the associations between gut microbiota and its plasma metabolites, with MS features. These authors identified a panoply of associations between gut microbiota composition and circulation metabolites, and MS features. For instance, these authors report an association between the microbiota metabolite trimethylamine N-oxide (TMAO, in the fasting plasma), associated with coronary artery disease and stroke, and the abundance of Peptococcaceae and *Prevotella*, and a negative association between TMAO and the abundance *F. prausnitzii*. These results underline that gut microbiota may modulate several cardio-metabolically traits [25].

Concerning microbiota metabolites, Xiaomin and colleagues [18] summarized current knowledge on the role of gut microbiota-derived tryptophan metabolites in the development of several diseases, including MS. Tryptophan is an essential amino acid, obtained from dietary proteins, and its metabolites, such as such as indole-3-lactate, indole-3-acrylate, indole-3-propionate, indole-3-aldehyde, indoleacetic acid, indole-3-acetaldehyde, and kynurenine (Kyn), can be produced by multiple taxa resident in the gut microbiota, and may have a role in MS pathogenesis. The metabolites can promote the differentiation and function of anti-inflammatory cells (such as anti-inflammatory macrophages and Treg cells) and are involved in maintaining the gut mucosal homeostasis [18]. Namely, blood levels of specific tryptophan metabolites are lower in patients with type 2 diabetes, when compared to the lean controls [17,26]. Also, a study using high fat fed rodents showed that increased acetate production, which occurs when microbiota is exposed to calorically dense nutrients, and particularly in the setting of chronic exposure to calorically dense food, promotes obesity and its related consequences of hyperlipidaemia, fatty liver disease, and insulin resistance [27].

On the other hand, Qin and colleagues [21] described that microbiota profile changes in patients with MS were associated with increased inflammation, through the inhibition of SCFAs production. A significantly lower microbiota diversity was observed in patients with MS. Namely, the relative abundance of Clostridiales (*Chlorobium phaeobacteroides, Clostridium asparagiforme, Clostridium bartlettii, Clostridium leptum, Clostridium scindens*, and *Collinsella aerofaciens*), five species from the order Bacteroidales (*Bacteroides fragilis, Roseburia intestinalis, Bacteroides nordii, Bacteroides thetaiotaomicron*, and *Bacteroides xylanisolven*), species from the genus *Alistipes* (*Alistipes onderdonkii, Alistipes hadrus, Alistipes colihominis*, and unclassified), and three species belonging to the family Ruminococcaceae (bacterium D16, *Ruminococcus lactaris*, and *Ruminococcus obeum*) were enriched in controls, when compared to MS patients. In addition, 28 bacterial species were negatively correlated with waist circumstance, being the strongest correlation with *Alistipes onderdonkii*. In line with these findings is the study from Vriezze and colleagues [28], in which microbiota transfer from lean donors to individuals with obesity and MS led to an increase in the abundance of butyrate-producing microbes and to an increase in insulin sensitivity six weeks after the procedure.

Given the data presented above, an association between microbiota and MS seems very likely and plausible. Despite the gap in knowledge regarding the specific microbiota profile in patients with MS, multiple data on modulation of microbiota in these patients is quickly arising.

## 3. Administration of Probiotic Supplements

### 3.1. Effects and Mechanisms of Action

Multiple factors associated to patients with MS, such as age and genetic background, cannot be changed, while other factors, such as weight and body mass index (BMI), triglycerides and high-density lipoprotein, or hypertension, can be somehow modifiable in order to improve the metabolic status of patients with MS [29]. Probiotics are alternatives which have been shown to be able to help to mitigate some of the described risk factors by enhancing the integrity of intestinal epithelium, adjusting inflammatory processes and endotoxin levels, modulating the bile acids production and secretion, and/or releasing antimicrobial peptides, among other mechanisms [30,31]. Therefore, it is important to know the mechanisms of action usually associated to the administration of probiotic supplements to the diet of patients with MS to understand and clarify its impact on metabolic health (Figure 1).

Improvements of the gut epithelial barrier, specifically among tight-junction proteins, can reduce bacterial translocation, inflammation, and metabolic endotoxaemia at the gut in patients with MS and these patients have been described with gut epithelium impairment [32,33,34]. Such gut impairment can be stimulated with poor diets and lack of certain nutrients. In the absence of fibers in the diet, the mucus barrier can work as source of nutrients for mucin-degrading bacteria, therefore affecting the epithelial thickness [35]. A firm inner structure associated to balanced microbiota, confers protection to the host [36]. *Lactobacillus reuteri* may compensate for impaired of aryl hydrocarbon receptors (related to some hormonal and immune responses) by increasing the availability of intestinal metabolites and improving metabolic homeostasis, being such results related to the restoration of the intestinal barrier function in animal models [37]. The Mediterranean diet, rich in polyunsaturated fats, polyphenols, carotenoids, and vitamins, was shown to be effective in reducing the risk of MS through the reinforcement of the gut barrier and the reduction of endotoxaemia in patients with in non-alcoholic fatty liver disease [38].

The most popular probiotics are members of lactobacilli and bifidobacteria groups, which are capable of interfering with dysbiotic gut biodiversity [39]. A higher Bacteroidetes/Firmicutes ratio is important in the gut and multiple probiotics have been showing the ability to modulate and normalize such ratio in murine models, as well as the abundance of Proteobacteria [40,41]. Specific gut bacteria, such as *Bilophila wadsworthia*, can also worsen the host metabolism in patients with high fat diets, being directly and indirectly related to inflammation mechanisms [42]. The probiotic *Lactobacillus rhamnosus* CNCM I-3690 was capable of reducing *B. wadsworthia*-induced immune and metabolic impairment by limiting its proliferation in the gut, reducing inflammation, and reinforcing intestinal barrier. The administration of multiple probiotics can also increase anti-inflammatory bacteria, such as *Prevotella*, in murine models of hepatocellular carcinoma along with their metabolites (i.e., propionate), shifting the bacteria community to Bacteroidetes, *Prevotella* and *Oscillibacter,* in addition to promoting IL-10 signalling and inhibiting pro-inflammatory helper T cell secretion from the gut to the liver [43].

By increasing proinflammatory molecules, such as lipopolysaccharides (LPS), it can be speculated that endotoxaemia can be promoted and metabolic disorders induced, therefore increasing the body fat mass and other metabolic parameters in obese patients. These effects can be reduced by probiotics through the preservation of gut permeability interfering with endotoxin levels [44]. Probiotic supplementation in rats may increase fatty acid oxidation, correct energy metabolism, plasma glucose and insulin resistance, inhibit cholesterol synthesis, prevent bile salt recycling, and modulate proinflammatory cytokines, therefore improving functional integrity of liver through the reduction of lipid reabsorption at the intestine [45]. Plasma bile acids, such as glycocholic acid, glycoursodeoxycholic acid, taurohyodeoxycholic acid, and tauroursodeoxycholic acid, were reduced in overweight adults taking synbiotics, supporting the effects of dietary supplements on certain metabolic pathways [46]. SCFAs, such as acetate, propionate and butyrate, can be released during the degradation of dietary fibers and are responsible for activities on the intestinal epithelial barrier, the immune system and the gut microbiota, sometimes working as bacterial inhibitors and quorum-sensing signaling molecules to regulate bacterial cell density and biofilm formation [36]. Nevertheless, it is important to decipher the potential beneficial anti-obesogenic, hypocholesterolemic, antihypertensive, and antiinflammatory properties of SCFAs and other metabolites produced and released by bacteria [47]. 

There are multiple probiotic strains described in the literature as presenting interesting and potential impact on MS. For example, *L. rhamnosus* BFE5264 resulted in a significant reduction of the serum cholesterol level that was accompanied by changes in intestinal microbiota and the production of SCFA in animal models [41]. *Bacillus licheniformis* Zhengchangsheng^®^ significantly decreased body weight gain and fat accumulation, serum lipid profiles, and proinflammatory cytokine levels, and improved glucose and lipid metabolism in obese mice [48]. *Lactobacillus gerneri* BNR17 was shown to inhibit the secretion of adiponectin and serum leptin and reduce mesenteric adipose tissue mass and adipocyte size in obese mice [49]. *Lactobacillus pentosus* GSSK2 and *Lactobacillus plantarum* GS26A exhibited improved glucose tolerance, liver biomarkers, alleviated oxidative stress, and restored the histoarchitechture of adipose tissue, colon, and liver, compared with high fat diet animals [45]. *L. reuteri* ATCC treated mice gained significantly less body weight than the control mice [50] and another strain of *L. reuteri* increased the expression of Cpt1a (gene involved in fatty acid oxidation pathway) in obese mice, although the lipogenic genes in the liver of mice were not altered by the probiotics [50]. *L. rhamnosus* NCIMB 8010 and *Pediococcus acidilactici* NCIMB 8018 improved the viability of human hepatocellular carcinoma cell line HepG2, protected against apoptosis under normal and insulin resistance conditions and attenuated oxidative stress by improving mitochondrial metabolism and dynamics [51]. *Bifidobacterium* supplementation ameliorated visceral fat accumulation and insulin sensitivity of the metabolic syndrome in rats under high fat diet [52]. Among the next-generation probiotics, *A. muciniphila* and *F. prausnitzii* are also promising candidates, being their abundance found reduced in different intestinal disorders [53] and increased in patients with MS [54].

### 3.2. Probiotics in MS

The search for Clinical Trials and Randomized Controlled Trials was conducted on PUBMED/MEDLINE, considering eligible articles published in English, French, Spanish, or Portuguese between January 1990 and September 2022. The terms used were “metabolic syndrome” and “probiotics” or “synbiotics”. Figure 2 shows the diagram for the selection of sources included in this systematic review.

The prophylactic potential of isolated probiotics in patients with MS has been tested in randomized clinical trials, but the results are still scarce. The results can be promising for particular probiotics, but the initial trials were not enthusiastic. *Lactobacillus salivarius* Ls-33 was tested on a series of biomarkers related to inflammation in adolescents with obesity and MS and no differences were observed after 12 weeks of treatment regarding anthropometric evaluation, blood pressure (systolic and diastolic), fasting glucose and insulin, homeostasis model assessment of insulin resistance, C-peptide, cholesterol, high-density lipoprotein cholesterol, low-density lipoprotein cholesterol, triglyceride, free fatty acids, C-reactive protein, interleukin-6, tumour necrosis factor-α, or faecal calprotectin [55]. In addition, *Lactobacillus casei* Shirota was tested by multiple studies regarding its effects on gut permeability, microbiome biodiversity and metabolite production, presence of endotoxin and neutrophil function in MS. Gut permeability can be significantly increased in MS as described above, but the treatment with *L. casei* Shirota did not show different results between patient and control groups [56]. Bacteroidetes/Firmicutes ratio was significantly higher in healthy controls compared to patients with MS, but the gut microbiome was not influenced by the probiotic. In addition, the proteins zonulin and calprotectin, usually higher in patients with MS, was not modified by the probiotic [32]; TMAO was not affected by *L. casei* Shirota either [57]. The insulin sensitivity index significantly improved after 3 months of probiotic supplementation, but the values were not different from the controls, as well as the values for β-cell and endothelial functions, or the inflammation markers [56,58].

More recently, other probiotics showed more success in clinical trials. The individual strain *L. reuteri* V3401 was tested by Tenorio-Jiménez and colleagues [59,60] and, although the decrease of Bacteroidetes/Firmicutes ratio was not corrected in obese patients, a rise of Verrucomicrobia was observed in patients receiving the probiotic. In addition, interleukin-6 and soluble vascular cell adhesion molecule 1 diminished following the treatment with the probiotic. Nevertheless, no significant correlation was observed between Verrucomicrobia abundance, and any inflammatory biomarker and subsequent studies are needed to complement the observations. Microbes4U© is a pilot study performed in patients with prediabetes and MS conducted to evaluate the tolerance, safety, and feasibility of the Gram-negative bacterium *A. muciniphila*, ingested either alive or pasteurized for 12 weeks, as a next-generation probiotic [61]. Beneficial impacts were shown on anthropometric measurements, as well as on the lipid profile, glycaemic parameters, such as insulin resistance, hepatic profile, and endotoxaemia, possibly due to interference with amino acids metabolism especially of alanine and arginine.

Multispecies probiotics may be more effective than single strain on metabolic disorders. Kassaian and colleagues [62] tested the effects of multiple probiotics (freeze-dried *Lactobacillus acidophilus, Bifidobacterium bifidum, Bifidobacterium lactis*, and *Bifidobacterium longum* with maltodextrin as filler) and synbiotics (the previous probiotics plus inulin as prebiotic) in individuals with prediabetes and MS. A clear reduction of hyperglycaemia in the groups treated with probiotic and synbiotic, as well as a reduction in hypertension in the group treated with probiotic, were reported.

### 3.3. Synbiotics in MS

The potential benefit of prebiotics can be conjugated with probiotics to potentiate its effects and support its adaptation and growth in challenging gut environments. Multiple sets of synbiotics have been tested in patients with MS, and the results have been clearly positive as described above by the study of Kassaian and colleagues [62]. Additional studies have been published and the results are in accordance. 

Synbiotic capsules containing *L. casei, L. rhamnosus, L. acidophilus, Lactobacillus bulgaricus, B. longum, Bifidobacterium breve*, and *Streptococcus thermophiles*, plus the prebiotic short chain fructo-oligosaccharide were tested on patients with MS [63]. The synbiotic treatment significantly reduced fasting blood glucose in the MS group versus placebo, but no differences were observed in other metabolic factors, including insulin level, homeostatic model assessment for insulin resistance, homoeostatic model assessment-β, and insulin/glucagon ratio. In another study, 38 patients with MS were supplemented with either synbiotic capsules containing seven strains of friendly bacteria (*L. casei, L. rhamnosus, L. acidophilus, L. bulgaricus B. longum, B. breve,* and *S. thermophilus*) plus fructo-oligosaccharide or placebo and increased the efficacy of diet therapy and the management of insulin resistance, although no significant differences were observed in low-density lipoprotein (LDL) levels, waist circumference, BMI, metabolism, and energy intake between the groups [64].

More relevant differences were reported by Rabiei and colleagues [65] by testing seven probiotic strains (*L. casei, L. rhamnosus, L. acidophilus, L. bulgaricus, B. longum, B. breve,* and *S. thermophilus*), plus fructo-oligosaccharide as prebiotic in patients with MS. The synbiotic treatment improved the status of BMI, fasting blood sugar, insulin resistance, homeostatic model assessment for insulin resistance, glucagon-like peptide-1, and peptide YY in patients, and interestingly, the trend of weight loss in the synbiotic group was significant until the end of the study. Cicero and colleagues [66] also tested a synbiotic formula comprising of *L. plantarum* PBS067, *L. acidophilus* PBS066, and *L. reuteri* PBS072 with active prebiotics in elderly patients with MS (aged 65–80 years). Patients receiving synbiotics improved waist circumference and fasting plasma insulin, arterial pressure, total cholesterol, high-density lipoprotein cholesterol, non-high-density lipoprotein cholesterol, triglycerides, low-density lipoprotein cholesterol, high-sensitivity C-reactive protein, and tumour necrosis factor-α serum levels. Compared to placebo, the patients receiving synbiotic treatment improved visceral adiposity index and triglycerides either. The EQ-5D Visual Analogue Scale (VAS) questionnaire confirmed an increase of quality of life in patients treated with synbiotics.

### 3.4. Other Foods with Probiotics in MS

The probiotics can be added to other foods and supplements and its effects have also been described in multiple studies and trials. The beneficial effects of functional yogurt NY-YP901 supplemented with mixture of *S. thermophilus, L. acidophilus, Bifidobacterium infantis,* and extra-ingredients containing *B. breve* CBG-C2, *Enterococcus faecalis* FK-23, fibersol-2 and other compounds, was tested in patients with MS [67]. In the group consuming NY-YP901, improvements were observed in body weight, BMI, and low-density lipoprotein-cholesterol after 8 weeks. A fortified yogurt containing the starter cultures of *S. thermophiles* and *L. bulgaricus* enriched with *B. lactis* Bb-12 was tested in overweight and obese patients with MS under a caloric-restricted diet [68]. The fortified yogurt reduced the body fat mass, body fat percentage, waist circumference, homoeostasis model of assessment-insulin resistance, triglyceride concentration versus patients consuming low fat yogurt, and led to a significant increase in total 25-hydroxyvitamin D, high density lipoprotein-cholesterol and quantitative insulin sensitivity check index. A probiotic yogurt containing *L. acidophilus* La5 and *B. lactis* Bb12 was compared with a regular yogurt for 2 months in patients with MS and significant reduction in the blood glucose and vascular cell adhesion molecule-1 was observed [69]. The probiotic yogurt induced changes in plasminogen activator inhibitor-1, insulin, homoeostasis model of assessment-insulin resistance, and quantitative insulin sensitivity check index compared to baseline, as well as improved fasting blood glucose and some serum markers associated to the endothelial function.

The influence of fermented milk with *L. plantarum* was tested in postmenopausal women with MS and showed positive results regarding CV risk factors by decreasing total cholesterol levels and fasting glucose levels [70]. In another study, the daily ingestion of fermented milk with *B. lactis* HN019 was tested in patients with MS and showed significant reduction in BMI, total cholesterol, low-density lipoprotein, tumour necrosis factor-α, and interleukin-6 pro-inflammatory cytokines when compared to baseline and control group values [71]. 

Probiotic kefir, comprising *Lactococcus lactis subsp. lactis, Lactococcus lactis subsp. cremoris, Lactococcus lactis subsp. diacetylactis, Leuconostoc mesenteroides subsp. cremoris, Lactobacillus kefyr, Kluyveromyces marxianus*, and *Saccharomyces unisporus*, was tested on patients with MS [72]. A significant increase in serum apolipoprotein A1 concentrations was provided by kefir compared to milk consumption. The regular kefir consumption did not provide superior effects compared with milk consumption on anthropometrical measurements, glycaemic control, inflammatory parameters, or blood pressure. Another study showed a decrease in fasting blood glucose without a change in glycated haemoglobin concentration after kefir (with more than 30 species of bacteria and more than 12 species of yeast and fungi) was administrated to patients with MS [73].

## 4. Discussion

In humans, data is being concordant towards beneficial effects of probiotics on patients with MS especially concerning weight loss, despite the effect is not transversal to all patients as described above. There are three important points to take into account when studying probiotics and its impacts on health. First, the individualized response to the consumption of probiotics may be dependent on microbiome variations and the ability of the probiotic strain(s) to interact and modify the host gut microbiome [74]. The gut microbiome and its variability is one of the first variables that need to be monitored in clinical trials in order to correctly compare patients. Patients should be carefully grouped, not only based on similar clinical features, but also taking into account the variability of the human microbiome as the response to modulatory treatments can be discrepant. Second, the variability of metabolic responses found among bacterial strains can be vast. For example, the strains *L. rhamnosus* LGG and *L. rhamnosus* BFE5264 belong to the same species, yet these strains may impact the gut microbiome of murine models for MS very differently and result in distinct cholesterol reduction levels [41]. These results strongly emphasise the importance of strain-specificity and metabolic networks potentially available in each strain. Third, the features of one probiotic formulation should not be generalized to multiple probiotics. The colony forming counts, type of strains, ratio of strains or the manufacturing processes of one probiotic product should be carefully considered and studied individually [75]. 

Although the mechanism of some probiotics has been clearly described and its impacts studied, it may be possible to combine probiotics strains via the complementary of mechanisms of action, therefore putting them to work together to achieve healthy goals. The metabolic deterioration of liver can be associated with excessive accumulation of free fatty acids, exhaustive oxidative stress, cellular apoptosis and inflammation, impairment of some insulin pathways and lipotoxicity [76], and probiotics may act on these multiple points as described above. Alternative mechanisms of action have been described for other probiotics in animal models and considerable advances may be soon seen in this topic. For example, *L. plantarum* PCS 26 might act as a liver X receptor agonist and help to improve lipid profiles in hypercholesterolemic patients with complex diseases, such as MS [77]. More recently, synbiotic supplementation showed recovering of nitric oxide function associated to hypertension in rats under high fat diets and correction of systolic blood pressure [78] and this represents a new and additional mechanism of action to be targeted. *L. plantarum* strains may also be capable to stimulate hepatic and renal nuclear factor-erythroid 2-related factor 2 (Nrf2) expression in hyperlipidemic mice and alleviate MS [79]. 

Engineered strains represent a dynamic and interesting new option for probiotics with specific activities and targets. An engineered *L. reuteri* secreting interleukin -22 was developed based on the probiotic *L. reuteri* ATCC PTA 6475 and could ameliorate non-alcoholic fatty liver disease [80]. Treatment with *L. reuteri* expressing interleukin-22 yielded subtle changes in the expression of reg3 genes in the small intestine and interleukin-22 levels in the plasma in some animal models. Ongoing research projects aimed to identify specific bacterial targets in the gut microbiome and then create phage cocktails designed to eliminate particular bacterial strains are also underway [36] and may represent a valid alternative for clinical cases associated to the proliferation of specific bacteria.

In this review it was described the effect of some probiotics and synbiotics currently available that were tested on patients with MS. Current results are very promising. In addition, it was observed that multiple strains (synbiotics) may be presenting better results on patients with MS due to the multitude of mechanisms of action that be working together in such cases. The number of trials available is still limited and the number of tested patients in each trial (some dozens) is also reduced. The ethnicity and nutritional habits tend to be similar as most of the studies were conducted in occidental countries, therefore, some differences may be observed when other populations are tested.

## 5. Conclusions

Although the specific microorganism profile in patients with MS is not yet properly known, these patients seem to have a different microbiota composition, when compared to patients without MS. Despite the gap in knowledge regarding the specific microbiota profile in patients with MS, multiple data on modulation of microbiota in these patients is quickly arising. It has been clearly described differences in the gut microbiome of patients with MS compared with healthy individuals, and such differences can be mitigated in some patients by the administration of probiotics or synbiotics. The number of published studies is still limited, and additional results can be expected soon as multiple randomized studies are currently being conducted. Nevertheless, there are multiple factors capable to affect the microbiota of patients with MS that should be considered simultaneously. Therefore, it is extremely difficult to associate particular microbial and metabolic changes to single factors. As more studies are published and both the diversity and stability of gut microbiome is revealed in patients with MS, a clear picture of the intricate relationship between microbiome and disease can become clear and additional therapeutic options can be explored.

## Figures and Tables

**Figure 1 nutrients-14-04490-f001:**
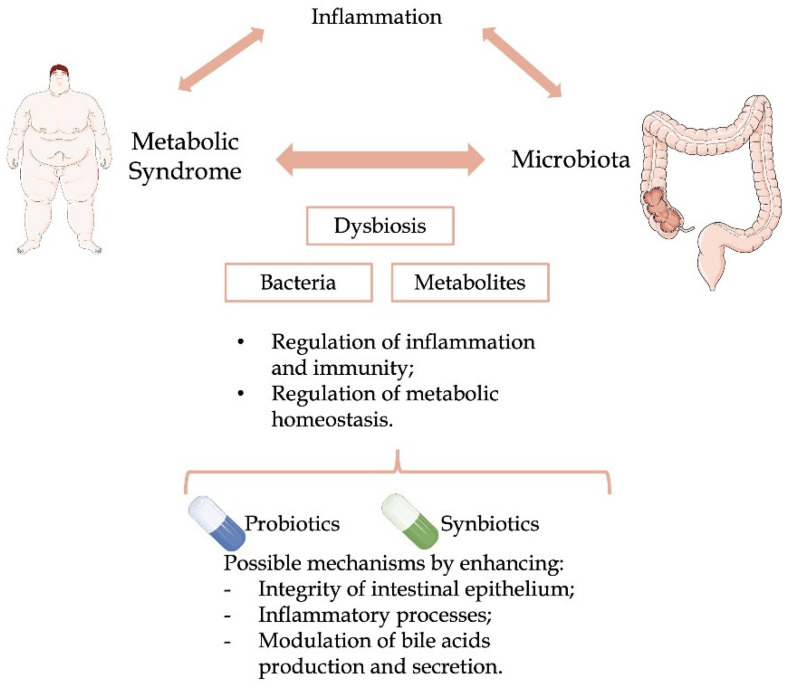
Mechanisms and modulation of the relationship between metabolic syndrome, human microbiome, and inflammation.

**Figure 2 nutrients-14-04490-f002:**
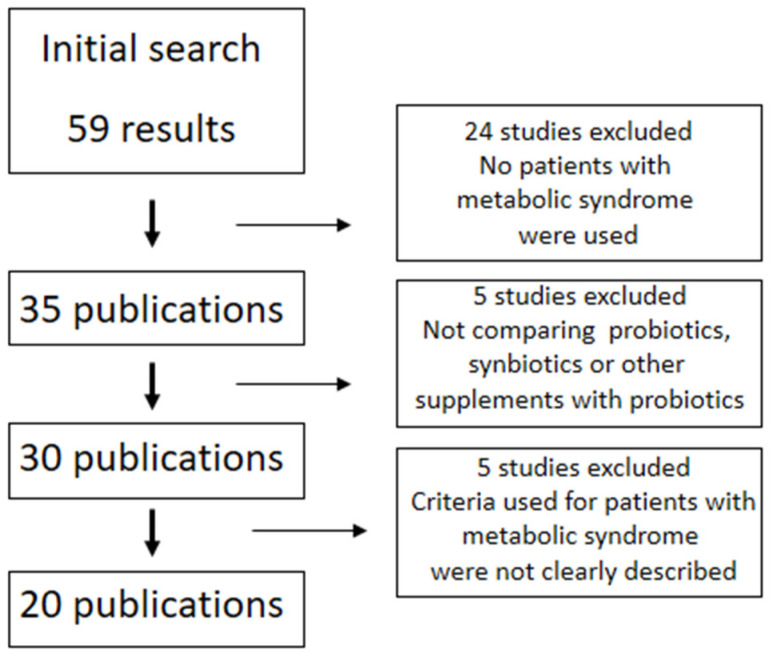
Diagram with the search results and criteria for selection of sources.

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
