# Peer review of "Microbiota Modulation in Patients with Metabolic Syndrome"

_nutrients, 2022, doi:10.3390/nu14214490_

Round 1
Reviewer 1 Report
The title of the article is chosen appropriately.
Abstract: Summarizes the article adequately. It doesn't need to be shortened.
Keywords: Sufficient number; does not need to be added or subtracted.
Introduction: A clear introduction is written without overwhelming the references.
The topics mentioned in the aim of the article have been adequately researched and summarized.
Discussion: There was a discussion that could be considered appropriate for a review article. The available references were evaluated; an achievable conclusion is written.
References: 80 references were used. The date range of the references is between 2004-2022 and is sufficient and up-to-date.
Author Response
Thanks for the comments and feedback.
Reviewer 2 Report
An interesting review, although I have doubts about the selection of sources. Most of the sources are older than 10 years, which should be indicated in the text (if we use historical sources, so as not to mislead the reader).
I ask the authors to complete the following:
- purpose and hypotheses of the review;
- methodology of the review (system and method of finding sources, inclusion and exclusion criteria, keywords, rationale for selection of sources; I recommend using methodology according to PRISMA Scope protocol.
- Include a graph showing the selection of sources;
- strengths and limitations of the review.
Author Response
An interesting review, although I have doubts about the selection of sources. Most of the sources are older than 10 years, which should be indicated in the text (if we use historical sources, so as not to mislead the reader).
Response: Thanks for the comments. We would like to notice that 40 out of 80 refs used in this review are from the last 5 years (2018-2022). Only 12 refs were published before 2011 and these refs represent important guidelines, definitions and research works relevant to the topic.
I ask the authors to complete the following:
- purpose and hypotheses of the review;
Response: The main objectives of the review were clarified and added to the manuscript. The purpose of the review was mainly:
1) To clarify the relationship between MS and gut microbiome
2) To describe the mechanism of action for potential microbial modulators
3) To perform a systematic review of the probiotics and synbiotics tested in patients with MS.
- methodology of the review (system and method of finding sources, inclusion and exclusion criteria, keywords, rationale for selection of sources; I recommend using methodology according to PRISMA Scope protocol.
Response: PRISMA Scope protocol was developed for systematic reviews and meta-analysis. This manuscript was designed to explore and described the link between MS and gut microbiome, as well as the potential impact of modulators. In the last part of the review we revised all the randomized clinical trials testing probiotics and synbiotics and the PRISMA Scope or other protocol can in fact be applied to this last part of the review. Such information was added to 3.2 section. The final 20 studies were used for sections 3.2, 3.3 and 3.4.
- Include a graph showing the selection of sources;
Response: Figure 2 was added with the information on the sources regarding the clinical trials and randomized clinical trials testing probiotics and synbiotics in patients with MS.
- strengths and limitations of the review.
Response: The information on strengths and limitations was added in the discussion (4.) section.
Round 2
Reviewer 2 Report
The authors revised the manuscript robustly for my recommendations. Thank you very much to the Authors and I recommend that the Publisher publish the paper.